**Data Availability Statement:** Minimal data set has been added as Supporting Information file. Some data cannot be shared publicly because of Ethical

# Anakinra in hospitalized COVID-19 patients guided by baseline soluble urokinase plasminogen receptor plasma levels: A real world, retrospective cohort study

**Francesco Vladimiro Segala**[1]◉*, **Emanuele Rando**[1]◉, **Federica Salvati**[1], **Marcantonio Negri**[2], **Francesca Catania**[1], **Flavia Sanmartin**[1], **Rita Murri**[1,3], **Evangelos J. Giamarellos-Bourboulis**[4], **Massimo Fantoni**[1,3]

**1** Dipartimento di Sicurezza e Bioetica—Sezione di Malattie Infettive, Università Cattolica del Sacro Cuore, Rome, Italy, **2** Dipartimento di Scienze Mediche e Chirurgiche, Fondazione Policlinico Universitario A. Gemelli IRCCS, Rome, Italy, **3** Dipartimento di Scienze di Laboratorio e Infettivologiche, Fondazione Policlinico Universitario A. Gemelli IRCCS, Rome, Italy, **4** 4th Department of Internal Medicine, National and Kapodistrian University of Athens, Medical School, Athens, Greece

◉ These authors contributed equally to this work.
* fvsegala@gmail.com

## Abstract

### Background

In patients with COVID-19 and baseline soluble urokinase plasminogen receptor plasma (suPAR) levels $\geq$ 6ng/mL, early administration of anakinra, a recombinant interleukin-1 receptor antagonist, may prevent disease progression and death. In case of suPAR testing unavailability, the Severe COvid Prediction Estimate (SCOPE) score may be used as an alternative in guiding treatment decisions.

### Methods

We conducted a monocenter, retrospective cohort study, including patients with SARS-CoV2 infection and respiratory failure. Patients treated with anakinra (anakinra group, AG) were compared to two control groups of patients who did not receive anakinra, respectively with $\geq$ 6 ng/mL (CG1) and < 6 ng/mL (CG2) baseline suPAR levels. Controls were manually paired by age, sex, date of admission and vaccination status and, for patients with high baseline suPAR, propensity score weighting for receiving anakinra was applied. Primary endpoint of the study was disease progression at day 14 from admission, as defined by patient distribution on a simplified version of the 11-point World Health Organization Clinical Progression Scale (WHO-CPS).

### Results

Between July, 2021 and January, 2022, 153 patients were included, among which 56 were treated with off-label anakinra, 49 retrospectively fulfilled prescriptive criteria for anakinra and were assigned to CG1, and 48 presented with suPAR levels < 6ng/mL and were

Committee requirements. Data are available from the Policlinico Gemelli Institutional Data Access (contact via Silvia Lamonica, silvia. lamonica@policlinicogemelli.it) for researchers who meet the criteria for access to confidential data.

**Funding:** The author(s) received no specific funding for this work.

**Competing interests:** I have read the journal's policy and the authors of this manuscript have the following competing interests: E. J. Giamarellos-Bourboulis has received honoraria from Abbott CH, bioMérieux, Brahms GmbH, GSK, InflaRx GmbH, Menarini and Sobi; independent educational grants from Abbott CH, AbbVie, bioMérieux Inc, Johnson & Johnson, MSD, UCB and Sobi and funding from the Horizon 2020 European Grants ImmunoSep and RISKinCOVID (granted to the Hellenic Institute for the Study of Sepsis) and the Horizn Health Grant EPIC-CROWN-2 (granted to the Hellenic Institute for the Study of Sepsis). M. Fantoni has received honoraria from Astra-Zeneca, GSK, Menarini, Sobi and funding from the Horizon 2020 European Grants (Covinform Project).

assigned to CG2. At day 14, when comparing to CG1, patients who received anakinra had significantly reduced odds of progressing towards worse clinical outcome both in ordinal regression analysis (OR 0.25, 95% CI 0.11–0.54, p<0.001) and in propensity-adjusted multiple logistic regression analysis (OR 0.32, 95% CI 0.12–0.82, p = 0.021) thus controlling for a wide number of covariates. Sensitivities of baseline suPAR and SCOPE score in predicting progression towards severe disease or death at day 14 were similar (83% vs 100%, p = 0.59).

## Conclusion

This real-word, retrospective cohort study confirmed the safety and the efficacy of suPAR-guided, early use of anakinra in hospitalized COVID-19 patients with respiratory failure.

## Background

Since it was first described in Wuhan, COVID-19 has resulted in an unprecedented health crisis, leading to 500 million reported cases [1] and to an estimated 18 million excess deaths [2]. Clinical presentation of SARS-CoV2 infection encompasses a large spectrum of manifestations, ranging from asymptomatic or mild, influenza-like symptoms, to pneumonia, severe respiratory failure and death [3]. Respiratory manifestations have often a sudden onset and are accompanied by systemic effects, indicating that SARS-CoV2 induces a major dysregulation of host response with a wide range of immuno-inflammatory alterations [4]. Thus, early identification of patients at risk and administration of timely, targeted treatments are crucial.

In a recent experimental study in mice, it was described that the injection of plasma from patients with severe COVID-19, enriched with danger-associated molecular patterns (DAMPs) like calprotectin (S100A8/A9), is inducing a compartmentalized inflammation in the host with the lungs and the intestine as the main sites of hyper-inflammation. Calprotectin was inducing the production of interleukin (IL)-1β. In the same model, the inhibition of murine interleukin (IL)-1α attenuated both pulmonary and intestinal inflammation [5]. Soluble urokinase plasminogen activator receptor was the biomarker which could predict the early increase of these DAMPs in the circulation release into its soluble form (suPAR) [6]. As a result, suPAR levels increase early in the disease process.

The potential role of suPAR in predicting severe respiratory failure (defined as partial oxygen pressure (PaO2) / fraction of inspired oxygen (FIO2) below 150 mmHg) and need for mechanical ventilation and death was suggested by Rovina et al. in 2020 [7] and, thereafter, two clinical trials were conducted to show the efficacy of early IL-1 targeting in COVID-19 patients presenting with suPAR levels above 6 ng/mL. The first is an open-label, non-randomized phase 2 clinical trial, the SAVE study, an interim analysis of which was published in March 2021 [8]. The second, the SAVE-MORE study, was a confirmatory, double-blind clinical trial, and they both showed significant efficacy of anakinra, a recombinant IL-1 receptor antagonist, in decreasing the risk of disease progression in patients with baseline suPAR levels ≥ 6 ng/mL [9]. In addition, to overcome the challenges posed by unavailability and low familiarity of physicians towards suPAR in clinical practice, the same authors developed and validated a clinical score based on baseline D-dimer, IL-6, CRP and ferritin levels, the SCOPE score [10]. The findings of SAVE and SAVE-MORE studies were then reviewed by the European Medicine Agency and led to the approval of biomarker-guided use of anakinra for the

treatment of COVID-19 pneumonia [11]. However, to our knowledge, apart from the case of SCOPE score validation, the efficacy of anakinra guided by baseline plasma suPAR levels has not been further explored. Also, patients recruited in SAVE and SAVE-MORE clinical trials were largely non-vaccinated against SARS-CoV2.

Aim of this study is to explore suPAR guided-use and efficacy of anakinra for the treatment of COVID-19 (100mg subcutaneously once daily for up to 10 days) and compare the performances of suPAR and the SCOPE score in a real world setting in a large University Hospital in Rome, between June 2021 and January 2022.

## Methods

### Study population and design

We conducted a retrospective cohort study at the University Hospital "Policlinico Agostino Gemelli IRCCS", Rome. All adult patients hospitalized for COVID-19 and treated with subcutaneous anakinra between the 1st July, 2021 and the 31rd January, 2022 were included. All patients were diagnosed with COVID-19 pneumonia using both real time-PCR and chest CT-scan.

As per institutional protocol, patients were eligible to receive anakinra if they presented with baseline suPAR levels $\geq$ 6 ng/mL, had respiratory failure (defined as PaO2/FiO2 ratio below 300 mmHg) requiring low-flow oxygen therapy (namely Venturi mask and nasal cannula), and did not show signs of neutropenia or severe bacterial co-infection. Baseline was defined as the date of hospital admission. Then, for each patient treated with anakinra ("anakinra group"), two controls were selected. The first control group (CG1) consisted of patients who retrospectively fulfilled the prescriptive criteria for anakinra but did not receive it at the discretion of the attending physicians, while the second control group (CG2) consisted of patients who developed COVID-related pneumonia but presented with baseline suPAR < 6ng/mL, and therefore did not fulfill the prescriptive criteria. Patients included into both control groups were manually matched to a given patient of the anakinra group according to the following criteria: date of admission (± 1 week), age (± 5 years), vaccine status against COVID-19 (whether they were fully vaccinated or not), and gender. Data collection, group allocation and matching were fully retrospective. A patient was defined to be fully vaccinated two weeks after receiving all recommended doses in the primary series of their COVID-19 vaccination [12]. We excluded, from all groups, patients diagnosed with moderate to severe immunodeficiency as defined by the CDC [13], subjects who presented with severe respiratory failure requiring high-flow oxygen therapy or the intensive care unit (ICU) on admission, patients included in clinical trials and individuals who received anti-IL6 treatment, namely tocilizumab or sarilumab.

All patients provided written informed consent to participate in the study and, in the period between study start and the approval of anakinra by the Italian Medicine Agency for the treatment of COVID-19 (23rd September, 2021), written informed consent for off-label use of anakinra was obtained to all individuals who received the study drug. The study was approved by "Fondazione Policlinico Gemelli IRCCS" ethical committee (protocol number 0010006/22, study ID 4770).

### Procedures and outcomes

All patients included in the study received standard of care for COVID-19 as per NIH guidelines [14]. Anakinra was administered subcutaneously at the dose of 100mg once a day, and the treatment was continued for up to ten days, according to patient response. Onset of significant adverse events, such as severe bacterial infection, granulocytopenia or acute elevation of

liver enzymes was also considered as an indication for treatment discontinuation. As part of routine laboratory exams requested for all COVID-19 patients, D-dimer, IL-6, Ferritin and CRP levels were measured, along with suPAR, on the day of hospitalization, and SCOPE score was calculated. Bacterial co-infection was defined as infection being diagnosed within the first 48 hours of hospital admission for COVID-19. When diagnosis occurred ≥ 48 hours after admission for COVID-19, these infections were defined as bacterial superinfections.

The main outcome of the study was disease progression at day 14 from admission. This was assessed using a simplified version of the WHO Clinical Progression Scale (WHO-CPS) [15]: a score of 1 was given for uninfected/ambulatory disease (WHO-CPS score 0–3), 2 for moderate disease (WHO-CPS score 4–5), 3 for severe disease (WHO-CPS score 6–9) and a score of 4 in case of death (WHO-CPS score 10). The same score was used by SAVE and SAVE-MORE studies, making the results of this real-world retrospective study more comparable to the ones of clinical trials.

Secondary outcomes were crude in-hospital mortality rate, length of hospitalization, development of severe respiratory failure with PaO2/FiO2 below 100 and 150 mmHg, and incidence of anakinra-related adverse events.

## Statistical analysis

The analysis aimed to investigate clinical and laboratory characteristics by comparing patients included in the anakinra group with patients belonging to the two control groups, as follows: patients treated with anakinra vs. patients with suPAR levels ≥ 6 ng/mL who were not treated with anakinra (CG1), and patients treated with anakinra vs. patients with suPAR levels < 6 ng/mL (CG2). Continuous variables were described using median and interquartile ranges, and categorical variables using frequencies and percentages. Wilcoxon rank-sum test was used to compare continuous variables and Pearson's χ2 test for categorical variables. A p-value of <0.05 was used to consider differences statistically significant. Since the p-value was potentially affected by small sample sizes, standardized differences (SD) were calculated by dividing the difference between the groups by the pooled standard deviation of the two groups (see S3 Table). An SD > 0.1 was interpreted as a meaningful difference. A propensity score (PS) of receiving anakinra was estimated through the use of a generalized boosted model. Covariates to include in the PS were identified by selecting variables with an SD > 0.1 in the comparison between patients with suPAR ≥ 6 ng/mL who were treated with anakinra, and patients not treated with anakinra (CG1). Variables with SD > 0.1 included in the PS were: age, smoker status, coronary artery disease, cerebrovascular disease, chronic kidney disease, PaO2/FiO2 ratio, C-reactive protein levels, white blood cells count, ferritin levels, D-dimer levels, use of dexamethasone, use of remdesivir, and the co-presence of a bacterial infection. A patient who was treated with anakinra was weighted by the inverse of the probability that he or she would be treated with anakinra, and a patient who did not receive anakinra was weighted by the inverse of the probability that he or she would not receive anakinra, equivalent to 1 minus his or her propensity score. The balance of the propensity model was later evaluated by verifying the obtained balance of PS covariates if they had an SD < 0.1 (see S1 Fig) and by comparing the baseline characteristics of the two exposure groups after applying the inverse probability of treatment weighting (IPTW). After that, crude and propensity-weighted single and multiple logistic regression models were performed to evaluate risk factors independently associated with the modified-WHO progression scale. Variables in the multiple logistic regression were restricted to only three due to respect the numerosity of outcomes and they were included if they had an influence on the primary outcome based on clinical importance by investigators' consensus. Variables included in the model were: anakinra use, age, and PaO2/FiO2 ratio.

Odds ratios and 95% confidence intervals (CI) were calculated. Multicollinearity was assessed by computing the variance inflation factor. Model predictive performances were assessed by calculating the ROC curve and the R2.

In the population of patients not treated with anakinra (CG1 and CG2), baseline suPAR and SCOPE score performances in predicting progression to severe disease and death were analyzed. Sensitivity, specificity, positive predictive value (PPV) and negative predictive value (NPV) were calculated by a 2 x 2 table. Diagnostic odds ratio and positive and negative likelihood ratios (LR+, LR−) were also estimated. Pearson's χ2 tests was run to assess heterogeneity of sensitivities and specificities between the two predictors, the null hypothesis being in both cases that all are equal.

Statistical analyses were performed with R software version 4.0.5 and RStudio version 1.4.1106 [16].

## Results

Between July 1, 2021 and January 31, 2022, 153 patients were included. Among them, 56 were treated with off-label anakinra, 49 retrospectively fulfilled prescriptive criteria for anakinra and were assigned to CG1, and 48 presented with suPAR levels < 6ng/mL and were assigned to CG2. Baseline characteristics of the overall population and of the three study groups and are reported in Table 1, while outcomes are reported in Table 2.

Overall, median age was 67 years (interquartile range, IQR, 55–77), 65% of enrolled subjects were males and median BMI was 26.6 (IQR, 24–30). Forty-three percent of included subjects were fully vaccinated. Patients belonging to the three group presented no significant differences in age, BMI, sex, vaccine status, time since symptom onset or comorbidities (with the exception of stroke). Median PaO2/FiO2 ratio at baseline was 250 mmHg (IQR 206–283). When comparing anakinra group with CG1, there was no significant difference in baseline levels of CRP, ferritin, IL-6, white blood cell count and SCOPE score. Also, even though severe bacterial co-infection was a contraindication to receive anakinra, no significant difference was noted between the two groups. P-values from the comparison between anakinra group and the CG1 are reported in Tables 1 and 2, while direct comparison between anakinra group and CG2 is reported in the appendix (pp 2–3). Median duration of oxygen therapy and hospital stay were, respectively, 10 (IQR, 7–17) and 11 days (IQR, 8–19).

Overall, 94% of study patients received dexamethasone, 32% received remdesivir, 7% tocilizumab and 6% received monoclonal antibodies against COVID-19. Median duration of anakinra was 6 days (IQR, 5–8). The drug was discontinued due to adverse event in two cases, once for liver enzymes elevation and once for development of pancytopenia. Incidence of bacterial co-infection was similar among the three groups.

When compared to CG1, patients in the anakinra group experienced less days of supplementary oxygen (10 vs 14 days, p-value = 0.032), had shorter hospital stay (11 vs 15 days, p-value = 0.008), and were less likely to progress to severe respiratory failure with PaO2/FiO2 < 150mmHg (23% vs. 47%, p-value = 0.011) during hospitalization. The results of the crude logistic regression model are reported in Table 3. As further shown in Table 4, receiving anakinra was associated with significantly reduced odds for progressing towards severe disease or death in the propensity-adjusted multiple logistic regression analysis (OR 0.32, p-value = 0.021), along with age (OR 1.09, p-value<0.001). However, no differences were recorded in HFNC use (29% vs. 41%, p-value = 0.19), ICU admission (3.6% vs. 4.1%, p-value>0.99) and death (7.1% vs. 12%, p-value = 0.51). Length of hospital stay and time to progression towards ARDS (defines as PF < 150 and use of HFNC) are shown in Fig 1, while day 14 allocation in the WHO-CPS are shown in Fig 2.

**Table 1. Patient's characteristics.**

| | Overall (n = 153) | suPAR < 6ng/mL | suPAR ≥ 6 ng/mL | | |
| --- | --- | --- | --- | --- | --- |
| | | Control Group 2[b] (n = 48) | Control Group 1[a] (n = 49) | Anakinra group (n = 56) | |
| | | | | | p-value[c] |
| Age, years, median (quartiles) | 67 (55–77) | 63 (50–74) | 68 (60–79) | 67 (57–78) | 0.54 |
| Male sex, n (%) | 99/153 (64.7) | 28/48 (58) | 32/49 (65) | 39/56 (70) | 0.64 |
| Fully vaccinated, n (%) | 62/146 (42.8) | 20/45 (44) | 20/45 (44) | 22/55 (40) | 0.65 |
| BMI, median (quartiles) | 26.6 (24.1, 30.3) | 27.2 (22.6–31.7) | 26.8 (24.5–30.3) | 26.2 (24.7–28.7) | 0.91 |
| Comorbidities, n (%) | | | | | |
| Smoking | 12/153 (7.8) | 3/48 (6.2) | 2/49 (4.1) | 7/56 (12) | 0.17 |
| COPD | 13/153 (8.5) | 4/48 (8.3) | 4/49 (8.2) | 5/56 (8.9) | >0.99 |
| High blood pressure | 72/153 (47.1) | 21/48 (44) | 24/49 (49) | 27/56 (48) | 0.94 |
| Coronary artery disease | 24/153 (15.8) | 1/48 (2.1) | 9/48 (19) | 14/56 (25) | 0.44 |
| Congestive heart failure | 7/153 (4.6) | 1/48 (2.1) | 3/49 (6.1) | 3/56 (5.4) | >0.99 |
| Atrial fibrillation | 12/153 (7.8) | 2/48 (4.2) | 4/49 (8.2) | 6/56 (11) | 0.75 |
| Stroke | 7/153 (4.6) | 3/48 (6.2) | 4/49 (8.2) | 0/56 (0) | **0.044** |
| Diabetes mellitus | 29/153 (18.9) | 6/48 (12) | 10/49 (20) | 12/56 (21) | >0.99 |
| Chronic kidney disease | 10 (6.5) | 3/48 (6.2) | 2/49 (4.1) | 5/56 (8.9) | 0.44 |
| Symptom onset-admission, days, median (quartiles) | 6.0 (3.0, 9.0) | 6.5 (5.0, 9.0) | 5.0 (2.0, 8.0) | 7.0 (3.8, 9.0) | 0.18 |
| Bacterial co-infection, n (%) | 7/153 (4.6) | 1/48 (2.1) | 5/49 (10) | 1/56 (1.8) | 0.10 |
| Bacterial supernfection, n (%) | 21 (13.7) | 5/48 (10) | 7/49 (14) | 9/56 (16) | 0.80 |
| PaO2/FiO2 ratio at baseline, mmHg, median (IQR) | 250 (206–283) | 274 (241–295) | 240 (196–269) | 240 (198–272) | 0.81 |
| Laboratory values at baseline, median (quartiles) | | | | | |
| Lymphocytes, cells/mm3 | 1010 (750–1395) | 1165 (880–1480) | 940 (740–1460) | 910 (730–1200) | 0.77 |
| WBC, cells/mm3 | 6440 (4735–8585) | 6335 (4195–9612) | 7030 (5045–8808) | 6200 (5005–8110) | 0.72 |
| IL-6, ng/L | 18 (8–46) | 13 (7–29) | 20 (8–59) | 32 (8–60) | 0.89 |
| Ferritin, ng/mL | 637 (276–1379) | 530 (253–1080) | 964 (375–1570) | 515 (246–1321) | 0.12 |
| D-dimer, ng/mL | 757 (534–1317) | 580 (367–1293) | 854 (618–1394) | 790 (575–1300) | 0.66 |
| CRP, mg/L | 71 (36–116) | 60 (20–94) | 106 (39–156) | 68 (39–124) | 0.26 |
| SCOPE score | 8 (6–9) | 6 (5–8) | 8 (6.3–10) | 8 (6.3–9) | 0.47 |
| suPAR, ng/mL | 6.9 (5.3–9.2) | 4.8 (4.1–5.3) | 9 (7–11.6) | 8 (6.8–9.8) | 0.14 |
| In-hospital therapy, n (%) | | | | | |
| Anakinra duration (days) | | - | - | 6 (5, 8) | - |
| Dexamethasone | 143/153 (94.1) | 41/47 (87) | 46/49 (94) | 56/56 (100) | 0.10 |
| Remdesivir | 49/153 (32) | 19/47 (40) | 12/49 (24) | 17/56 (30) | 0.58 |
| Tocilizumab | 11/153 (7.2) | 2/47 (4.3) | 9/49 (18) | 0/56 (0) | **<0.001** |
| Monoclonal antibodies | 9/153 (5.9) | 6/47 (13) | 2/49 (4.1) | 1/56 (1.8) | 0.60 |

COPD: chronic obstructive pulmonary disease; ICU: intensive-care unit; HFNC: high-flow nasal cannula

[a] patients who retrospectively fulfilled the prescriptive criteria for anakinra but did not receive the drug.

[b] patients admitted for COVID-19 pneumonia but who presented with baseline suPAR < 6 ng/mL.

[c] p-values from the comparison between anakinra group and CG1.

When ordinal regression analysis was done comparing the allocation of the WHO-CPS strata by day 14 (Table 2), it was found that anakinra treatment was associated with 0.25 odds for less worse outcome than comparators (p-value <0.0001). This is fully corroborating the analysis of anakinra efficacy of the SAVE-MORE trial [8].

**Table 2. Outcomes.**

|  | Overall (n = 153) | suPAR < 6ng/mL | suPAR ≥ 6 ng/mL | |  |
|---|---|---|---|---|---|
|  |  | Control Group 2[b] (n = 48) | Control Group 1[a] (n = 49) | Anakinra group (n = 56) | *p*-value[c] |
| Supplementary oxygen, days, median (quartiles) | 10 (7, 17) | 7 (5, 11) | 14 (9, 20) | 10 (8, 15) | 0.032 |
| Length of stay, days, median (quartiles) | 11 (8, 19) | 10 (7, 14) | 15 (10, 23) | 11 (8, 18) | **0.008** |
| Need of high flow oxygen between days 1 and 14 n (%) | 42/153 (27.5) | 6/48 (12) | 20/49 (41) | 16/56 (29) | 0.19 |
| PaO2/FiO2 < 100, n (%) | 17/153 (11.1) | 1/48 (2.1) | 9/49 (18) | 7/56 (12) | 0.40 |
| PaO2/FiO2 < 150, n (%) | 40/153 (26.1) | 4/48 (8.3) | 23/49 (47) | 13/56 (23) | **0.011** |
| Admission to ICU, n (%) | 4/153 (2.6) | 0/48 (0) | 2/49 (4.1) | 2/56 (3.6) | >0.99 |
| Non-invasive ventilation, n (%) | 3/153 (2) | 0/48 (0) | 2/49 (4.1) | 1/56 (1.8) | 0.60 |
| Mechanical ventilation, n (%) | 1 (0.7) | 0/48 (0) | 0/49 (0) | 1/56 (1.8) | >0.99 |
| Allocation to strata of the WHO Clinical Progression Scale at day 14, n (%) | | | | | Odds ratio 0.25 (0.11–0.54) p<0.0001[d] |
| 0–3 | 87/149 (58.4) | 36/47 (77) | 17/49 (35) | 34/54 (64) | |
| 4–5 | 47/149 (3.5) | 9/47 (19) | 22/49 (45) | 16/54 (30) | |
| 6–9 | 9/149 (6) | 2/47 (4.3) | 5/49 (10) | 2/54 (3.8) | |
| 10 | 6/149 (4) | 0/47 (0) | 5/49 (10) | 2/54 (1.9) | |
| In-hospital death, n (%) | 11/153 (7.2) | 1/48 (2.1) | 6/49 (12) | 4/56 (7.1) | 0.51 |

[a] patients who retrospectively fulfilled the prescriptive criteria for anakinra but did not receive the drug.

[b] patients admitted for COVID-19 pneumonia but who presented with baseline suPAR < 6 ng/mL.

[c] p-values from the comparison between anakinra group and CG1.

[d] Ordinal regression analysis comparing anakinra group and CG1.

Performances of baseline suPAR and SCOPE score in predicting progression towards severe disease or death at day 14 from admission are reported in Fig 3. Overall, sensitivities were similar (83% vs 100%, *p*-value = 0.59) but suPAR was more specific (54% vs 29%, *p*-value = 0.003). In this study, SCOPE score negative predictive value was 100%.

## Discussion

Our study confirmed the clinical benefit of administration of subcutaneous anakinra in preventing disease progression in patients hospitalized for COVID-19 pneumonia with elevated suPAR levels at the time of admission. To our knowledge, this is the first study exploring performances of suPAR in identifying candidates to receive anakinra outside of a clinical trial. Here, patients treated with anakinra were compared to two control groups of patients who did

**Table 3. Crude multiple logistic regression for risk factors associated with progression towards severe disease or death at day 14.**

| Population: patients with baseline suPAR ≥ 6ng/mL who fulfilled the prescriptive criteria for anakinra (n = 105) | | | | | | |
|---|---|---|---|---|---|---|
| Variable | Univariable | | | Multivariable | | |
|  | OR[1] | 95% CI[1] | *p*-value | OR[1] | 95% CI[1] | *p*-value |
| Age (years) | 1.07 | 1.03, 1.13 | 0.005 | 1.09 | 1.03, 1.15 | 0.004 |
| Anakinra | 0.31 | 0.08, 1.01 | 0.064 | 0.32 | 0.08, 1.12 | 0.087 |
| PaO2/FiO2 ratio | 1.00 | 0.99, 1.01 | 0.371 | 0.99 | 0.98, 1.00 | 0.154 |

OR: Odds Ratio; CI: Confidence Interval.

Severe disease is defined according to the WHO Clinical Progression Scale (WHO-CPS score 6–10).

**Table 4. Propensity-adjusted multiple logistic regression for risk factors associated with progression towards severe disease or death at day 14.**

Population: patients with baseline suPAR ≥ 6ng/mL who fulfilled the prescriptive criteria for anakinra (n = 105)

| Variable | Univariable | | | Multivariable | | |
|---|---|---|---|---|---|---|
| | OR[1] | 95% CI[1] | *p*-value | OR[1] | 95% CI[1] | *p*-value |
| Age (years) | 1.07 | 1.04, 1.12 | < 0.001 | 1.09 | 1.05, 1.14 | < 0.001 |
| Anakinra | 0.34 | 0.13, 0.79 | 0.016 | 0.32 | 0.12, 0.82 | 0.021 |
| PaO$_2$/FiO$_2$ ratio | 1.00 | 0.99, 1.00 | 0.243 | 0.99 | 0.98, 1.00 | 0.051 |

OR: Odds Ratio; CI: Confidence Interval.

Severe disease is defined according to the WHO Clinical Progression Scale (WHO-CPS score 6–10).

not receive anakinra, respectively with ≥ 6 ng/mL and < 6 ng/mL baseline suPAR. This was done to further explore the role of this biomarker in early patient identification.

Among the 105 subjects with elevated suPAR levels at admission, 56 patients were treated with off-label anakinra and showed significantly lower odds of worsening to severe disease and death by day 14, and these results were confirmed even when controlling for age and PaO2/FiO2 ratio. Disease severity was defined in accordance with the WHO clinical progression scale, a composite outcome that defines "severe disease" as progression towards respiratory failure requiring high-flow oxygen therapy and/or non-invasive and/or invasive mechanical

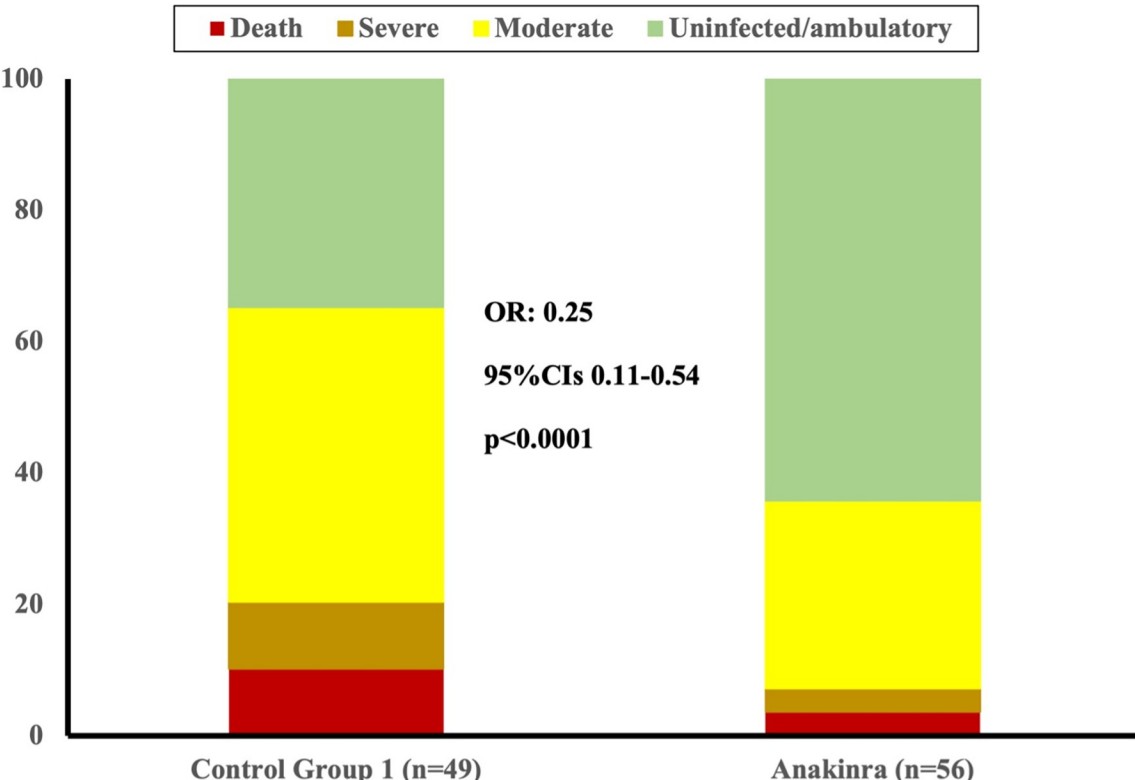

**Fig 1. Time to progression towards ARDS and length of hospital stay of patients with baseline SuPAR ≥ 6 ng/mL who received and did not receive anakinra.** Time to progression to ARDS (A) and length of hospital stay (B) of patients admitted for COVID-19 with baseline SuPAR ≥ 6 ng/mL and non-severe respiratory failure who received and did not receive anakinra. ARDS was defined as PaO2/FiO2 < 150 mmHg and use of high-flow oxygen therapy/NIV or MV. COVID-19 = coronavirus disease 2019. ARDS = acute respiratory distress syndrome. SuPAR = soluble urokinase plasminogen activator receptor. HR = hazard ratio.

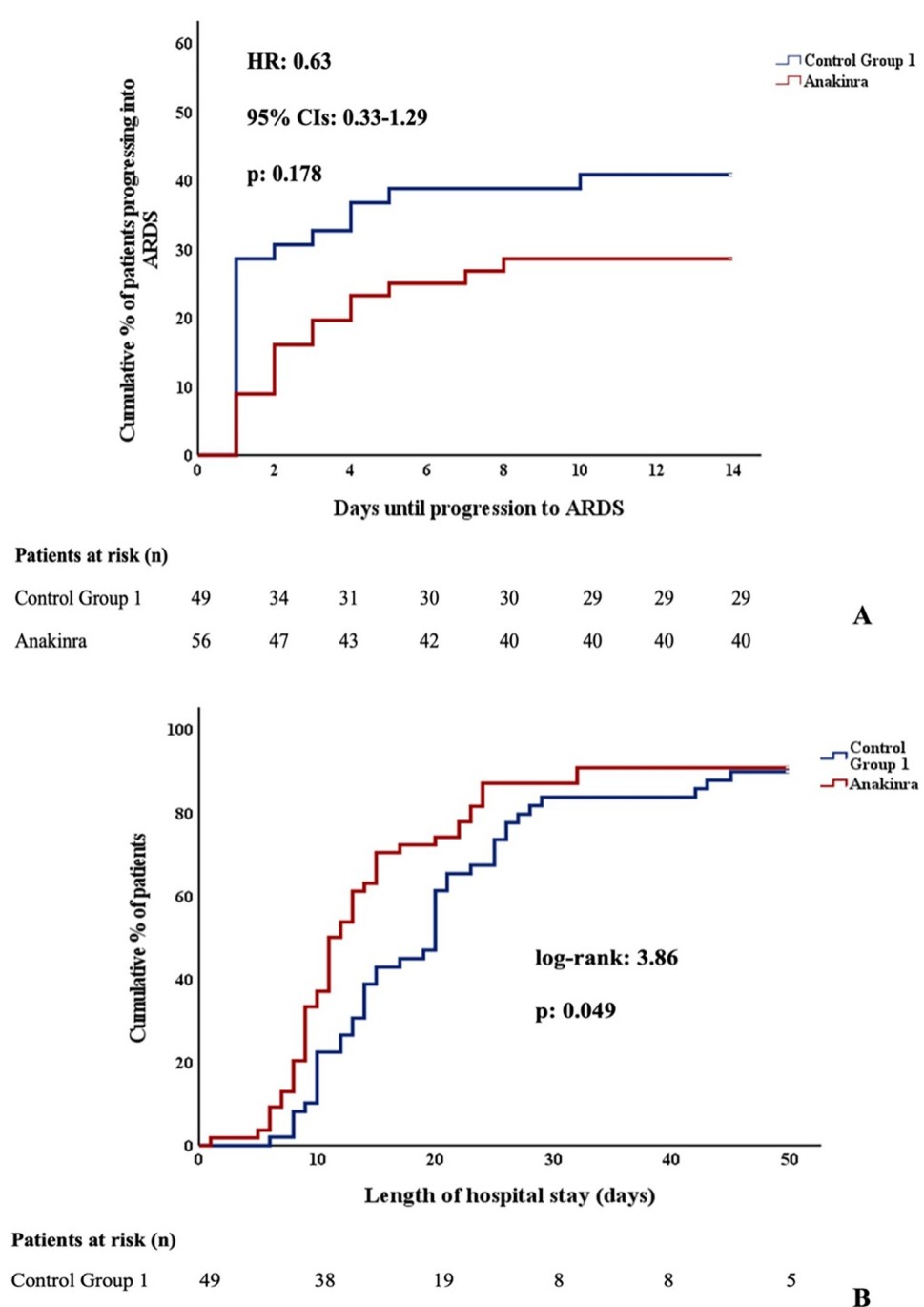

**Fig 2. Disease progression at day 14 from hospital admission.** Distribution of the WHO-CPS scores at day 28 of patients admitted for COVID-19 with baseline SuPAR ≥ 6 ng/mL and non-severe respiratory failure who received and did not receive anakinra (Control Group 1). Comparison is done by unadjusted ordinal regression analysis; the ORs of the 95% CIs are provided. COVID-19 = coronavirus disease 2019. SuPAR = soluble urokinase plasminogen activator receptor. OR = odds ratio.

ventilation at a certain time point [15]. In addition, individuals who received anakinra experienced a 4-day shorter median hospital stay and were less likely to evolve towards severe respiratory failure with PaO2 / FiO2 below 150mmHg during hospitalization. Importantly, patients

| | Severe disease/death | | |
|---|---|---|---|
| | (+) | (-) | total |
| suPAR ≥ 6 (n) | 10 | 39 | 49 |
| suPAR < 6 (n) | 2 | 45 | 47 |
| total (n) | 12 | 84 | 96 |

| | Severe disease/death | | |
|---|---|---|---|
| | (+) | (-) | total |
| SCOPE ≥ 6 (n) | 11 | 55 | 66 |
| SCOPE < 6 (n) | 0 | 23 | 23 |
| total (n) | 11 | 78 | 89 |

| | suPAR < 6 | SCOPE ≥ 6 | p-value |
|---|---|---|---|
| sensitivity | 83% | 100% | 0.588 |
| specificity | 54% | 29% | 0.003 |
| Positive predictive value | 20% | 17% | |
| Negative predictive value | 96% | 100% | |
| Diagnostic odds ratio (95% CI) | 4.83 (1.1-20.5) | 9.7 (0.5-172) | |
| Positive LR (95% CI) | 1.7 (1.2-2.4) | 1.4 (1.1-1.6) | |
| Negative LR (95% CI) | 0.36 (0.1-1.1) | 0.14 (0.01-2.1) | |

population: patients not treated with anakinra (n=96)

**Fig 3. suPAR vs SCOPE score performances in predicting progression to severe disease or death at day 14.** Severe disease is defined according to the WHO Clinical Progression Scale (WHO-CPS score 6–10). CI = Confidence Interval; LR = likelihood ratio.

belonging to both control groups were paired for date of admission and were selected to have similar age, sex and rate of vaccination against SARS-CoV2 to patients treated with anakinra (Table 1). Coupling for date of admission was done to avoid differences in standard of care (SoC) and circulating COVID variants, since Delta and Omicron variants were associated with varying disease severity [17] and published data from clinical trials were obtained from patients hospitalized before both variants became dominant in Europe [18, 19]. Also, data from clinical trials did not control the benefit of anakinra for patients fully vaccinated against SARS-CoV2, as mass vaccination campaigns initiated few months before the end of SAVE-MORE enrollment.

In this study, there were no significant differences in the rate of vaccinated and non-vaccinated individuals and, among patients with elevated suPAR, the magnitude of the efficacy of anakinra in preventing disease progression was maintained regardless to vaccine status. Being vaccinated was in fact associated with a trend towards higher odds of disease progression, but this may be biased by the fact that vaccinated people were older and generally presented more risk factors for disease progression than non-vaccinated people (appendix, pg 4). Interestingly, along decreased odds for disease progression, non-vaccinated people presented with a lower baseline PaO2/FiO2 and increased inflammatory markers, thus suggesting that anakinra benefit may be more pronounced in this population.

Furthermore, among patients with elevated suPAR baseline levels, there was no statistical difference in terms of prescription of dexamethasone, remdesivir, and anti-SARS-CoV2 monoclonal antibodies. By contrast, when compared to patients treated with anakinra, individuals with low baseline suPAR showed a more favorable baseline profile and evolved towards less severe infections (appendix, pp 2–3).

A peculiar finding of the present study is that median duration of anakinra was 6 days (IQR, 5–8), in contrast with SAVE-MORE study protocol, which called for a treatment duration of 10 days and in any case not inferior to 7 days [20]. Being a retrospective, real-world study, a minimum treatment duration was not defined, leaving the decision whether to keep treating or discontinue to the attending clinician. Therefore, apart from two cases (3.6%), in which potentially drug-related adverse events occurred, anakinra was interrupted due to clinical benefit, namely the resolution of respiratory failure and withdrawal of oxygen supplementation.

The prescriptive heterogeneity reported here is a consequence of the fact that some of the prescribers in charge for the patients participated to SAVE-MORE clinical trial, and the clinical

experience gained during trial recruitment influenced their familiarity with both suPAR predictive accuracy and anakinra safety profile. Thus, it is likely that this experience led to the acquisition of the confidence necessary to keep prescribing anakinra also in their daily practice, outside of the clinical trial, as an off-label medication. Indeed, only clinicians who participated to the SAVE-MORE trial became familiar with suPAR interpretation, since the measurement of this biomarker was implemented in our center only at the beginning of the study period. Thus, at the time of prescription, there was no intent to allocate patients to any "intervention" or "control" group. There were no specific clinical reasons why CG1 patients did not receive the study drug and data collection, group allocation and matching were entirely retrospective. Written informed consent for administration of any off-label medication was required by our institution.

These findings support the assumption that a biomarker-guided, early prescriptive strategy of an immunomodulatory drug for the treatment of an acute infection is still challenging to be implemented, even in a setting where the biomarker is readily available. However, recognition of the importance of the role of biomarkers in early patient identification is critical in guiding IL-1 blockade, since several studies, both randomized [21, 22] and observational [23, 24], reported conflicting results–as well as a Cochrane systematic review [25]–likely as a consequence of the fact that IL-1 antagonists prescribed at a disease stage that was already too advanced for the drug to provide substantial benefit. Yet, poor availability of suPAR assays is a major limitation for this approach, as non-widespread suPAR availability was cited by NIH guidelines to be the key factor preventing anakinra to enter among the Panel recommended drugs [26]. In this respect, this study provides further evidence to support the role of the SCOPE score as an alternative to baseline suPAR. This was done in line with the assumption, consistent with the work of Giamarellos-Bourboulis et al. [10], that SCOPE score may serve as an alternative marker of early activation of the pathological processes that foster COVID-19 disease progression, namely the inflammatory, endothelial and coagulation pathways. Indeed, our retrospective analysis confirmed that suPAR and SCOPE score have similar sensitivities in predicting severe disease and death at day 14. Importantly, in individuals who were not treated with anakinra, SCOPE score showed a negative predictive value of 100%, thus supporting its potential value in guiding prescriptive decision making.

Our study has obvious limitations. First, being retrospective and observational in nature, its results may be biased by confounding factors that could have affected clinical progression apart from the study drug. However, baseline characteristics among the three groups, and especially among anakinra group and patients with baseline suPAR $\geq$ 6ng/mL (CG1), were similar. Furthermore, as discussed above, this risk was minimized by selecting controls according to age, vaccine status and gender. Second, observational studies carry the risk of immortal time bias [27], but this bias has been minimized by the fact that we excluded patients admitted or considered for admission in ICU. Also, even if they have intrinsic limitations, observational studies have the advantage to provide data from real-world scenario, without the selection bias that may affect perspective studies. Third, it is important to acknowledge that the results may have been biased by a low sample size, even if the results are consistent with the findings of clinical trials. Fourth, the main study outcome did not differentiate between asymptomatic and mildly symptomatic disease, as well as, among patients categorized as developing "severe disease", between the ones treated with HFNC, mechanical ventilation, vasopressors, dialysis and ECMO (WHO-CPS score 6 to 9).

## Conclusions

In conclusion, this real-word, retrospective cohort study confirmed the safety and the efficacy of suPAR-guided, early use of anakinra in hospitalized COVID-19 patients with respiratory

failure. Our results support the indication that all COVID patients with suPAR levels ≥ 6ng/ml at time of hospitalization should receive anakinra, including the ones belonging to the control group of this study.

Also, it provides further evidence in support of the utilization of SCOPE score in guiding clinical decision-making as an alternative to suPAR. However, despite its approval by drug regulatory agencies, further evidence is needed for the widespread dissemination of this personalized, biomarker-guided therapeutic approach to COVID-19.

## Supporting information

**S1 Fig. Covariates balance after applying the generalized boosted model to estimate the propensity score of receiving anakinra.**
(DOCX)

**S1 Table. Patients with suPAR < 6 ng/mL vs patients treated with anakinra.**
(DOCX)

**S2 Table. Fully vaccinated vs non vaccinated individuals, descriptive statistics.**
(DOCX)

**S3 Table. Standardized mean differences (SMD) comparing patients with suPAR≥ 6 ng/mL treated with anakinra (1) and not treated with anakinra (0).**
(DOCX)

**S1 Dataset.**
(XLSX)

## Acknowledgments

We thank Francesca Schinzari, Annalisa Potenza (Policlinico "A. Gemelli" IRCCS, Malattie Infettive Columbus) and all the other fellow first-line colleagues of the "COVID Columbus" hospital who, despite working incessantly and passionately to contain COVID-19 pandemic, also found time to provide support for the development of this report.

## Author Contributions

**Conceptualization:** Francesco Vladimiro Segala, Federica Salvati, Marcantonio Negri, Rita Murri, Evangelos J. Giamarellos-Bourboulis, Massimo Fantoni.

**Data curation:** Francesco Vladimiro Segala, Emanuele Rando, Federica Salvati, Francesca Catania, Flavia Sanmartin.

**Formal analysis:** Francesco Vladimiro Segala, Emanuele Rando, Evangelos J. Giamarellos-Bourboulis.

**Investigation:** Francesco Vladimiro Segala.

**Methodology:** Francesco Vladimiro Segala.

**Project administration:** Francesco Vladimiro Segala, Marcantonio Negri, Massimo Fantoni.

**Supervision:** Rita Murri, Evangelos J. Giamarellos-Bourboulis, Massimo Fantoni.

**Validation:** Rita Murri, Evangelos J. Giamarellos-Bourboulis, Massimo Fantoni.

**Visualization:** Francesco Vladimiro Segala, Marcantonio Negri, Rita Murri, Evangelos J. Giamarellos-Bourboulis, Massimo Fantoni.

**Writing – original draft:** Francesco Vladimiro Segala, Emanuele Rando, Evangelos J. Giamarellos-Bourboulis.

**Writing – review & editing:** Francesco Vladimiro Segala.

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
