## [Decision Letter · Decision Letter 0]

18 Oct 2022

PONE-D-22-21590Anakinra in hospitalized COVID-19 patients guided by baseline soluble urokinase plasminogen receptor plasma levels: a real world, retrospective cohort studyPLOS ONE

Dear Dr. Segala,

Thank you for submitting your manuscript to PLOS ONE. After careful consideration, we feel that it has merit but does not fully meet PLOS ONE’s publication criteria as it currently stands. Therefore, we invite you to submit a revised version of the manuscript that addresses the points raised during the review process.

We look forward to receiving your revised manuscript.

Kind regards,

Cecilia Acuti Martellucci, M.D.

Academic Editor

PLOS ONE

Journal Requirements:

2. In the Methods section of your revised manuscript, please amend lines 112-116 to match the description in lines 298-299 (Written informed consent for administration of any off-label medication was required by our institution.).

Reviewers' comments:

Reviewer's Responses to Questions

**Comments to the Author**

1. Is the manuscript technically sound, and do the data support the conclusions?

Reviewer #1: Yes

Reviewer #2: Yes

2. Has the statistical analysis been performed appropriately and rigorously? 

Reviewer #1: No

Reviewer #2: Yes

3. Have the authors made all data underlying the findings in their manuscript fully available?

Reviewer #1: Yes

Reviewer #2: No

4. Is the manuscript presented in an intelligible fashion and written in standard English?

Reviewer #1: Yes

Reviewer #2: Yes

5. Review Comments to the Author

Reviewer #1: REVIEW PONE

The Authors present a retrospective study evaluating the real-life use of subcutaneous anakinra (ANK) in patients with COVID-19 pneumonia. In the paper, three groups are described: patients with suPAR > 6ng/mL who received ANK, patients with suPAR <6ng/mL who did not receive ANK and patients with suPAR <6ng/mL. The Authors show that the use of ANK reduced the odds of progression toward worse clinical outcomes and hospital length of stay. The paper is well written and provides an interesting real-life experience on the use of ANK.

However, a few major points should be addressed:

- Line 97: the Authors should explain their definition of “severe” when defining bacterial co-infections;

- Line 110-111: it is not clear if patients who received anti-IL6 treatments were excluded or not, as in the paper 7% received tocilizumab; given that patients with immunosuppression were excluded, it does not seem that the Authors refer to previous treatment with anti-IL6 molecules for other condition; this issue should be clarified;

- Line 136: why did the Authors analyze two different cut-offs for severe respiratory failure (100 and 150)? Given the endpoints and the definition of ARDS I would use only the 150 cut-off;

- Was suPAR requested for all patients admitted with COVID-19 during the study period? If not, what are the characteristics of the patients for whom suPAR was and was not requested?

- The selection of patients in the control groups does not appear to be clear. Were all patients with suPAR >6ng/mL included in the study, or were only patients used for the matching procedure considered? If so, how many patients presented with suPAR>6ng/mL, were not treated with ANK and were not included in the study? And what were the characteristics of these patients?

- In the statistical analysis section, more information should be provided regarding the matching procedure and the associated analysis (i.e., use of conditional logistic regression analysis, which is suggested for this kind of studies).

- The time from admission to administration of ANK should be described;

- The main outcome is disease progression at day 14, using a simplified version of the WHO-CPS; however, in Figure 1 the Authors evaluated progression to ARDS; I believe it would be more coherent with the main outcome to describe time to progression towards severe disease (which was defined as WHO-CPS 6-9), and possibly leave progression to ARDS as a secondary outcome; moreover, the definition of ARDS should be coherent throughout the text (Line 203-204: PF<150 and use of HFNC; line 216-217 PF <150 and use of HFNC/NIV or MV).

- Results and table 3: where are the results of the matching cohort described? It seems from table 3 that 105 patients are included, i.e. 49+56. If a 1:1 matching was performed, the number of patients should be different.

- Line 269-275: the association between vaccine status and outcome is not statistically significant at univariate and multivariate analysis, so I would delete the phrase “Being vaccinated was in fact associated with a trend towards higher odds of disease progression, but this may be biased by the fact that vaccinated people were older and generally presented more risk factors for disease progression than non-vaccinated people (appendix, pg 4).”, while I would keep the following one (Line 272-275).

- Line 324: how did you exclude patients considered for admission to ICU?

- Figure 1: The difference between the two groups appears more significant in the first 24-48h. It seems that about 30% of patients in the control group progressed to ARDS in the first 24h, compared to 10% of patients in the ANK group, so it would be important to know the time of administration of ANK (see also the previous comment)

Minor comments:

- Line 92: 31rd should be changed to 31st

- Line 194: “co-infections” should probably be changed to “superinfections”

- Line 203: “defines” should be “defined”;

- Secondary outcomes are “crude in-hospital mortality rate, length of hospitalization, development of severe respiratory failure with PaO2/FiO2 below 100 and 150 mmHg, and incidence of anakinra-related adverse events”; however, in the results section, supplementary oxygen and ICU admissions are also described. This should be added in the methods section.

- Line 230-233: I would specify that patients analyzed did not receive anakinra (i.e: “performances of baseline suPAR and SCOPE score in predicting progression towards severe disease or death at day 14 from admission in patients non receiving ANK…”)

- Line 320-321: baseline characteristics were not very similar between patients with suPAR >6 and <6, so I would just say that patients with suPAR >6 had similar characteristics.

Reviewer #2: This is a small retrospective real-world study about supar guided anakinra treatment in COVID-19. This is the first study of real-world data. My concerns are as follows:

-As the study was retrospective, please make clear how written informed consent was received from participants. This is not clearly described and easy to follow.

-Please describe in more detail the matching statistical procedure.

-In the manuscript the authors describe that tocilizumab was an exclusion criterion but in Table 1 some participants have received tocilizumab. Please explain

-Table 1 and 2: it would be easier for the reader to provide here also p values of comparison with CG2 instead of supplement

- Incidence of bacterial co-infection was similar: it would be very interesting to provide these data

-Lines 288-299: here the authors discuss something very important, namely the Hawthorne effect. It is very important to have real world data and see how a strategy developed in the context of a clinical trial finds applicability and is easily implemented in everyday routine. This is of great value in real world data studies such as this one, which is the first after the two clinical trials for suPAR guided anakinra.

6. PLOS authors have the option to publish the peer review history of their article (what does this mean?). If published, this will include your full peer review and any attached files.

Reviewer #1: No

Reviewer #2: **Yes: **Evdoxia Kyriazopoulou

---

## [Author Response · Author response to Decision Letter 0]

26 Nov 2022

Dear PLOS-One Editor,

please find attached a point-by-point response to Journal Requirements and to Reviewers.

Response to Journal Requirements

R: Style requirements have been checked on the provided links. Supplementary information have been renamed and title page has been reformatted.

2. In the Methods section of your revised manuscript, please amend lines 112-116 to match the description in lines 298-299 (Written informed consent for administration of any off-label medication was required by our institution).

R: To avoid redundancy and misunderstanding, the sentence “Written informed consent for administration of any off-label medication was required by our institution” has been removed. 

3. In your Data Availability statement, you have not specified where the minimal data set underlying the results described in your manuscript can be found.

R: Thank you for the remark. Minimal data set has been added as Supporting Information file.

4. Please include captions for your Supporting Information files at the end of your manuscript, and update any in-text citations to match accordingly.

R: File naming and in-text citations have been modified as per journal requirements.

Response to Reviewer #1

Major comments:

1. The Authors should explain their definition of “severe” when defining bacterial co-infections

R: Bacterial co-infection was considered “severe” when associated with life-threatening organ dysfunction (Singer et al. JAMA, 2016). Definition has been added in the text (Lines 105-106).

2. It is not clear if patients who received anti-IL6 treatments were excluded or not, as in the paper 7% received tocilizumab; given that patients with immunosuppression were excluded, it does not seem that the Authors refer to previous treatment with anti-IL6 molecules for other condition; this issue should be clarified

R: We thank the reviewer for the observation. Anti-IL6 treatment has been considered as an exclusion criterion for study enrollment as in our institutional protocol it was listed among the contraindications to anakinra administration. However, 18% (9/49) patients included in CG1 and 4.3% (2/47) patients included in CG2 received later during hospitalization, as part of the standard of treatment for COVID-19, accounting for 7.2% of the total population. In contrast, as shown in Table 1, patients who received anakinra did not receive anti-IL6 during the whole course of the hospitalization. The issue has been clarified in the text (Lines 121-124)

3. Why did the Authors analyze two different cut-offs for severe respiratory failure (100 and 150)? Given the endpoints and the definition of ARDS I would use only the 150 cut-off.

R: Thank you for the observation. We agree with the Reviewer, and we removed the 100mmHg cut-off.

4. Was suPAR requested for all patients admitted with COVID-19 during the study period? If not, what are the characteristics of the patients for whom suPAR was and was not requested?

R: Yes, once our laboratory implemented the suPAR assay, it was requested for every new COVID patient as part of the routine laboratory exams requested at admission.

5. The selection of patients in the control groups does not appear to be clear. Were all patients with suPAR >6ng/mL included in the study, or were only patients used for the matching procedure considered? If so, how many patients presented with suPAR>6ng/mL, were not treated with ANK and were not included in the study? And what were the characteristics of these patients? 

R: Thank you for the observation. Matching was done by the authors of this study by directly looking for a suitable patient on the internal informatic system. For every patient treated with anakinra, all collectors followed the same matching procedure:

1. Searched for patients admitted for COVID-19 in the same time period (± 1 week). 

2. Among patients admitted in the same time period, data collectors identified the ones with similar age (± 5 years), gender and vaccinal status. Those patients were then considered eligible for the study.

3. Screened for inclusion criteria.

4. Screened for exclusion criteria.

Hence, we did not collect data of all the other patients with baseline suPAR ≥ 6ng/ml, since the description of this population felt outside the scope of the present study. 

More details about the matching procedure have been added in the text (Lines 112-120)

6. In the statistical analysis section, more information should be provided regarding the matching procedure and the associated analysis (i.e., use of conditional logistic regression analysis, which is suggested for this kind of studies).

R: As reported in the previous response, matching was done entirely by the authors of the study (E.R., F.S., F.C, F.S. and F.V.S.), actively looking for eligible patients within the hospital informatic system and, therefore, statistical analysis was introduced only upon data collection completion.

7. The time from admission to administration of ANK should be described.

R: We thank the Reviewer for the observation. Median time from admission to ANK administration was 1day (IQR: 1-2). We added this data to Table 1.

8. The main outcome is disease progression at day 14, using a simplified version of the WHO-CPS; however, in Figure 1 the Authors evaluated progression to ARDS; I believe it would be more coherent with the main outcome to describe time to progression towards severe disease (which was defined as WHO-CPS 6-9), and possibly leave progression to ARDS as a secondary outcome; moreover, the definition of ARDS should be coherent throughout the text (Line 203-204: PF<150 and use of HFNC; line 216-217 PF <150 and use of HFNC/NIV or MV).

R: We agree with the Reviewer: progression towards severe disease is indeed the main outcome, and progression towards ARDS is one the secondary outcomes. We clarified this in the Methods section (Lines 161-164) and, to avoid confusion, we inverted the naming of Figure 1 and Figure 2. As suggested by the reviewer, ARDS definition has been uniformed throughout the text.

9. Results and table 3: where are the results of the matching cohort described? It seems from table 3 that 105 patients are included, i.e. 49+56. If a 1:1 matching was performed, the number of patients should be different.

R: As reported in Table 1, matching cohort 1 and 2 (CG1 and CG2) were composed, respectively, by 49 and 48 patients. Unfortunately, to respect eligibility and inclusion criteria for controls, we were not able to reach exact 1:1 matching.

10. The association between vaccine status and outcome is not statistically significant at univariate and multivariate analysis, so I would delete the phrase “Being vaccinated was in fact associated with a trend towards higher odds of disease progression, but this may be biased by the fact that vaccinated people were older and generally presented more risk factors for disease progression than non-vaccinated people (appendix, pg 4).”, while I would keep the following one (Line 272-275).

R: We are grateful to the Reviewer for the observation. The phrase was removed from the text.

11. How did you exclude patients considered for admission to ICU?

R: We excluded patients considered for admission in ICU as they did not fulfil inclusion and exclusion criteria: in our center, all patients considered for COVID-ICU presented with severe respiratory failure with PaO2/FiO2 < 150mmHg requiring either high-flow oxygen therapy or NIV/MV.

12. Figure 1: The difference between the two groups appears more significant in the first 24-48h. It seems that about 30% of patients in the control group progressed to ARDS in the first 24h, compared to 10% of patients in the ANK group, so it would be important to know the time of administration of ANK (see also the previous comment).

R: We warmly thank the reviewer for this observation. It is correct, and it matches with the time of ANK administration (median: 1 day after admission). A dedicated sentence has been added in the Discussion section (Lines 306-310).

Minor comments:

13. Line 92: 31rd should be changed to 31st

14. Line 194: “co-infections” should probably be changed to “superinfections”

15. Line 203: “defines” should be “defined”;

R: The above sentences have been corrected. 

16. Secondary outcomes are “crude in-hospital mortality rate, length of hospitalization, development of severe respiratory failure with PaO2/FiO2 below 100 and 150 mmHg, and incidence of anakinra-related adverse events”; however, in the results section, supplementary oxygen and ICU admissions are also described. This should be added in the methods section.

R: All secondary outcomes have been added in the Methods section (Lines 149-162).

17. Lines 230-233: I would specify that patients analyzed did not receive anakinra (i.e: “performances of baseline suPAR and SCOPE score in predicting progression towards severe disease or death at day 14 from admission in patients non receiving ANK…”)

18. Line 320-321: baseline characteristics were not very similar between patients with suPAR >6 and <6, so I would just say that patients with suPAR >6 had similar characteristics.

R: Thank you. We modified the sentences as suggested (Line 272 and 386).

Response to Reviewer #2

1. As the study was retrospective, please make clear how written informed consent was received from participants. This is not clearly described and easy to follow.

R: We thank the Reviewer for the observation. Along with (when necessary) asking for consent to off-label medications, in our center all COVID patients were asked for informed consent to use anonymized data for research purposes. This was a routine procedure in our hospital. This has been clarified in the methods section (Lines 126-127).

2. Please describe in more detail the matching statistical procedure.

R: Thank you for the observation. Matching was done by the authors of this study by directly looking for a suitable patient on the internal informatic system. For every patient treated with anakinra, all collectors followed the same matching procedure:

1. Searched for patients admitted for COVID-19 in the same time period (± 1 week). 

2. Among patients admitted in the same time period, data collectors identified the ones with similar age (± 5 years), gender and vaccinal status. Those patients were then considered eligible for the study.

3. Screened for inclusion criteria.

4. Screened for exclusion criteria. 

More details about the matching procedure have been added in the text (Lines 112-120).

3. In the manuscript the authors describe that tocilizumab was an exclusion criterion but in Table 1 some participants have received tocilizumab. Please explain.

R: We thank the reviewer for the observation. Anti-IL6 treatment has been considered as an exclusion criterion for study enrollment as in our institutional protocol it was listed among the contraindications to anakinra administration. However, 18% (9/49) patients included in CG1 and 4.3% (2/47) patients included in CG2 received later during hospitalization, as part of the standard of treatment for COVID-19, accounting for 7.2% of the total population. In contrast, as shown in Table 1, patients who received anakinra did not receive anti-IL6 during the whole course of the hospitalization. The issue has been clarified in the text (Lines 121-124).

4. Table 1 and 2: it would be easier for the reader to provide here also p values of comparison with CG2 instead of supplement.

R: We agree with the Reviewer. P-values obtained from the comparison of AG and CG2 have been moved from supplement to Table 1 and 2.

5. Incidence of bacterial co-infection was similar: it would be very interesting to provide these data.

R: The seven co-infections reported in the study were all presumptive bacterial pneumonia complicating COVID-19. Clinical suspicion raised mainly from CT-scan presentation and elevation in the polymorphonucleate count at admission. All patients were prescribed wide-spectrum empirical antimicrobial therapy. Unfortunately, none of the reported co-infections resulted positive to blood or sputum cultures.

6. Lines 288-299: here the authors discuss something very important, namely the Hawthorne effect. It is very important to have real world data and see how a strategy developed in the context of a clinical trial finds applicability and is easily implemented in everyday routine. This is of great value in real world data studies such as this one, which is the first after the two clinical trials for suPAR guided anakinra.

R: We warmly thank the Reviewer for this comment.

We are thankful for all the issues highlighted by reviewers. Hence, we believe that reviewer’s observations substantially improved the quality of our work.

Kind regards,

Francesco Vladimiro Segala

---

## [Decision Letter · Decision Letter 1]

2 Jan 2023

PONE-D-22-21590R1Anakinra in hospitalized COVID-19 patients guided by baseline soluble urokinase plasminogen receptor plasma levels: a real world, retrospective cohort studyPLOS ONE

Dear Dr. Segala,

Thank you for submitting your manuscript to PLOS ONE. After careful consideration, we feel that it has merit but does not fully meet PLOS ONE’s publication criteria as it currently stands. Therefore, we invite you to submit a revised version of the manuscript that addresses the points raised during the review process.

We look forward to receiving your revised manuscript.

Kind regards,

Cecilia Acuti Martellucci, M.D.

Academic Editor

PLOS ONE

Additional Editor Comments:

I thank the authors for the work done to improve the manuscript. Reviewer 2 has some further points that should be addressed before this paper can be considered for publication.

Reviewers' comments:

Reviewer's Responses to Questions

**Comments to the Author**

1. If the authors have adequately addressed your comments raised in a previous round of review and you feel that this manuscript is now acceptable for publication, you may indicate that here to bypass the “Comments to the Author” section, enter your conflict of interest statement in the “Confidential to Editor” section, and submit your "Accept" recommendation.

Reviewer #1: (No Response)

Reviewer #2: All comments have been addressed

2. Is the manuscript technically sound, and do the data support the conclusions?

Reviewer #1: Partly

Reviewer #2: (No Response)

3. Has the statistical analysis been performed appropriately and rigorously? 

Reviewer #1: No

Reviewer #2: (No Response)

4. Have the authors made all data underlying the findings in their manuscript fully available?

Reviewer #1: Yes

Reviewer #2: (No Response)

5. Is the manuscript presented in an intelligible fashion and written in standard English?

Reviewer #1: Yes

Reviewer #2: (No Response)

6. Review Comments to the Author

Reviewer #1: I commend the Authors of the article for adequately addressing most of the comments raised in the first round of review. However, I believe that some issues still have to be resolved regarding the statistical analysis. Specifically:

- For the analysis of the primary outcome, the matching between the CG1 and the anakinra group should be 1:1 to respect the assumption of a matched-cohort study; in order to reach a 1:1 matching, therefore, some patients of the anakinra group who do not have an exact matching should be excluded from the analysis;

- The Authors state that ordinal regression analysis was used to explore risk factors associated with progression to severe disease or death; however, if the ordinal scale ranging from 6 to 10 was used as dependent variable, some of the categories would have 0 patients (especially in the anakinra group, where there are 2 patients in the “severe disease” group, ranging from 6 to 9); it would be more adequate to perform a binomial logistic regression analysis using as dependent variable progression to severe disease or death vs. no progression to severe disease or death);

- Given the number of patients with the outcome of interest (14 patients, according to table 1), the number of variables used in the multivariate analysis appears too high, according to the commonly used rule-of-thumb; moreover, it is not clear how the variables were chosen to be included in the multivariate analysis (as, for example, BMI and sex were not associated with the outcome in the univariate analysis); it would be more prudent to reduce the number of variables included in the multivariate analysis, possibly by selecting variable with significant p-values at univariable analysis;

- The statistical methods for the primary outcome should then be better specified in the “Statistical analysis” section.

Minor comment:

- I would add the word “manually” in line 111 (...patients included in both control groups were manually matched…)

Reviewer #2: (No Response)

7. PLOS authors have the option to publish the peer review history of their article (what does this mean?). If published, this will include your full peer review and any attached files.

Reviewer #1: No

Reviewer #2: **Yes: **Evdoxia Kyriazopoulou

---

## [Author Response · Author response to Decision Letter 1]

23 Jan 2023

Major comments:

1. For the analysis of the primary outcome, the matching between the CG1 and the anakinra group should be 1:1 to respect the assumption of a mahed-cohort study; in order to reach a 1:1 matching, therefore, some patients of the anakinra group who do not have an exact matching should be excluded from the analysis

R: Thank you for your considerable observation. In order to improve the internal validity of our study, we performed the analysis with a more rigorous methodology. Indeed, although the two cohorts initially appeared to be well-balanced with respect to important baseline characteristics, subtle imbalance may still be present if only looking at the p-value due to a small sample size effect. Thus, we evaluated the standardized mean differences (SMD) of the covariates between the two groups; in this way, we found that several covariates were actually differently distributed. For this reason, we went further by creating a propensity score (PS) of receiving anakinra to avoid unbalanced distribution of variables. After that, we use the PS to estimate the inverse probability of treatment weighting (IPTW), conducting the analysis adjusting for these weights. By such a method, we were able to use all patients, regardless of exactly matching 1:1, by assigning each patient a weight [1]. 

2. The Authors state that ordinal regression analysis was used to explore risk factors associated with progression to severe disease or death; however, if the ordinal scale ranging from 6 to 10 was used as dependent variable, some of the categories would have 0 patients (especially in the anakinra group, where there are 2 patients in the “severe disease” group, ranging from 6 to 9); it would be more adequate to perform a binomial logistic regression analysis using as dependent variable progression to severe disease or death vs. no progression to severe disease or death)

R: Thank you for the observation. For ordinal logistic regression, we used a simplified version of the simplified version of the 11-point World Health Organization Clinical Progression Scale, ranging from 1 (uninfected/ambulatory disease) to 4 (death). Severe disease (WHO-CPS score 6-9) are categorized with a score of 3 in the simplified version (lines 129-134, methods section), thus all patients are categorized. Unfortunately, due to the retrospective nature of our data, we were not able to use the full WHO-CPS scale. We agree with the reviewer that this represents a limitation, and we added a paragraph to highlight this in the dedicated section (lines 407-409). However, even using a simplified scale, with the aim to be more comparable with the SAVE-MORE results, the authors of this work consider ordinal regression analysis as a good tool to represent our results.

3. Given the number of patients with the outcome of interest (14 patients, according to table 1), the number of variables used in the multivariate analysis appears too high, according to the commonly used rule-of-thumb; moreover, it is not clear how the variables were chosen to be included in the multivariate analysis (as, for example, BMI and sex were not associated with the outcome in the univariate analysis); it would be more prudent to reduce the number of variables included in the multivariate analysis, possibly by selecting variable with significant p-values at univariable analysis.

R: Thank you very much for the precious observation. As stated in the response to the first comment, we modified the analysis using a propensity score of receiving anakinra along with the IPTW. We selected the variables to include in the PS by looking if the standardized mean differences of the covariates between the two groups were > 0.1 since this cutoff is generally considered a meaningful difference. Concerning the variables to include in the propensity-adjusted multiple logistic regression, we follow the rule of thumb as suggested to improve the validity of the results. Indeed, we included only 3 variables: anakinra use, age in years, and PaO2/FiO2 ratio. Those variables were selected by investigators’ consensus and clinical importance. We used these variables due to their profound impact on the outcome in order to evaluate the role of anakinra independently of them. Moreover, even if some covariates were not perfectly balanced in the PS (WBC, CRP, and PaO2/FiO2 ratio), we included only the PaO2/FiO2 ratio to further adjust for it; with respect to the other two variables, we avoided including them due to their SD < 0.25 not deeply influencing the overall balance, and to limit the number of predictors. We preferred not to use variables below a certain cutoff of p due to some caveats associated with this method, similar to stepwise selection strategies [2]. Doing so, we argued that our model is more explanatory respecting its original purpose and avoiding possible misleading results due to possible false-positive p. 

4. The statistical methods for the primary outcome should then be better specified in the “Statistical analysis” section.

R: Thank you for your observation. We also agree with your considerations and rewrite the entire section as follows: 

“The analysis aimed to investigate clinical and laboratory characteristics by comparing patients included in the anakinra group with patients belonging to the two control groups, as follows: patients treated with anakinra vs. patients with suPAR levels ≥ 6 ng/mL who were not treated with anakinra (CG1), and patients treated with anakinra vs. patients with suPAR levels < 6 ng/mL (CG2). Continuous variables were described using median and interquartile ranges, and categorical variables using frequencies and percentages. Wilcoxon rank-sum test was used to compare continuous variables and Pearson’s χ2 test for categorical variables. A p-value of <0.05 was used to consider differences statistically significant. Since the p-value was potentially affected by small sample sizes, standardized differences (SD) were calculated by dividing the difference between the groups by the pooled standard deviation of the two groups. An SD > 0.1 was interpreted as a meaningful difference. A propensity score (PS) of receiving anakinra was estimated through the use of a generalized boosted model. Covariates to include in the PS were identified by selecting variables with an SD > 0.1 in the comparison between patients with suPAR ≥ 6 ng/mL who were treated with anakinra, and patients not treated with anakinra (CG1). Variables with SD > 0.1 included in the PS were: age, smoker status, coronary artery disease, cerebrovascular disease, chronic kidney disease, PaO2/FiO2 ratio, C-reactive protein levels, white blood cells count, ferritin levels, D-dimer levels, use of dexamethasone, use of remdesivir, and the co-presence of a bacterial infection. A patient who was treated with anakinra was weighted by the inverse of the probability that he or she would be treated with anakinra, and a patient who did not receive anakinra was weighted by the inverse of the probability that he or she would not receive anakinra, equivalent to 1 minus his or her propensity score. The balance of the propensity model was later evaluated by verifying the obtained balance of PS covariates and by comparing the baseline characteristics of the two exposure groups after applying the IPTW. After that, crude and propensity-weighted single and multiple logistic regression models were performed to evaluate risk factors independently associated with the modified-WHO progression scale. Variables in the multiple logistic regression were restricted to only three due to respect the numerosity of outcomes and they were included if they had an influence on the primary outcome based on clinical importance by investigators’ consensus. Variables included in the model were: anakinra use, age, and PaO2/FiO2 ratio. Odds ratios and 95% confidence intervals (CI) were calculated. Multicollinearity was assessed by computing the variance inflation factor. Model predictive performances were assessed by calculating the ROC curve and the R2. In the population of patients not treated with anakinra (CG1 and CG2), baseline suPAR and SCOPE score performances in predicting progression to severe disease and death were analyzed. Sensitivity, specificity, positive predictive value (PPV), and negative predictive value (NPV) were calculated by a 2 x 2 table. Diagnostic odds ratio and positive and negative likelihood ratios (LR+, LR−) were also estimated. Pearson’s χ2 tests were run to assess heterogeneity of sensitivities and specificities between the two predictors, the null hypothesis being in both cases that all are equal.”. 

Minor comments:

5. I would add the word “manually” in line 111 (...patients included in both control groups were manually matched…)

R: We added the word “manually” as suggested.

---

## [Decision Letter · Decision Letter 2]

23 Feb 2023

PONE-D-22-21590R2Anakinra in hospitalized COVID-19 patients guided by baseline soluble urokinase plasminogen receptor plasma levels: a real world, retrospective cohort studyPLOS ONE

Dear Dr. Segala,

Thank you for submitting your manuscript to PLOS ONE. After careful consideration, we feel that it has merit but does not fully meet PLOS ONE’s publication criteria as it currently stands. Therefore, we invite you to submit a revised version of the manuscript that addresses the points raised during the review process.

We look forward to receiving your revised manuscript.

Kind regards,

Cecilia Acuti Martellucci, M.D.

Academic Editor

PLOS ONE

Journal Requirements:

Additional Editor Comments (if provided):

The manuscript is acceptable for publication, however please first amend the abstract according to the suggestion by Reviewer 1.

Reviewers' comments:

Reviewer's Responses to Questions

**Comments to the Author**

1. If the authors have adequately addressed your comments raised in a previous round of review and you feel that this manuscript is now acceptable for publication, you may indicate that here to bypass the “Comments to the Author” section, enter your conflict of interest statement in the “Confidential to Editor” section, and submit your "Accept" recommendation.

Reviewer #1: All comments have been addressed

Reviewer #2: All comments have been addressed

2. Is the manuscript technically sound, and do the data support the conclusions?

Reviewer #1: Yes

Reviewer #2: (No Response)

3. Has the statistical analysis been performed appropriately and rigorously? 

Reviewer #1: Yes

Reviewer #2: (No Response)

4. Have the authors made all data underlying the findings in their manuscript fully available?

Reviewer #1: Yes

Reviewer #2: (No Response)

5. Is the manuscript presented in an intelligible fashion and written in standard English?

Reviewer #1: Yes

Reviewer #2: (No Response)

6. Review Comments to the Author

Reviewer #1: The Authors adequately addressed all the raised concerns.

The abstract should be modified according to the final version of the manuscript.

Reviewer #2: (No Response)

7. PLOS authors have the option to publish the peer review history of their article (what does this mean?). If published, this will include your full peer review and any attached files.

Reviewer #1: No

Reviewer #2: **Yes: **Evdoxia Kyriazopoulou

---

## [Author Response · Author response to Decision Letter 2]

28 Feb 2023

Response to Reviewer #1

The Authors adequately addressed all the raised concerns. The abstract should be modified according to the final version of the manuscript.

R: Thank you. Abstract was modified.

---

## [Editor Report · Decision Letter 3]

7 Mar 2023

Anakinra in hospitalized COVID-19 patients guided by baseline soluble urokinase plasminogen receptor plasma levels: a real world, retrospective cohort study

PONE-D-22-21590R3

Dear Dr. Segala,

We’re pleased to inform you that your manuscript has been judged scientifically suitable for publication and will be formally accepted for publication once it meets all outstanding technical requirements.

Kind regards,

Cecilia Acuti Martellucci, M.D.

Academic Editor

PLOS ONE

---

## [Editor Report · Acceptance letter]

23 Mar 2023

PONE-D-22-21590R3 

Anakinra in hospitalized COVID-19 patients guided by baseline soluble urokinase plasminogen receptor plasma levels: a real world, retrospective cohort study 

Dear Dr. Segala:

I'm pleased to inform you that your manuscript has been deemed suitable for publication in PLOS ONE. Congratulations! Your manuscript is now with our production department. 

Kind regards, 

on behalf of

Dr. Cecilia Acuti Martellucci 

Academic Editor

PLOS ONE